# Characterization of the Slovene Autochthonous Rose Hybrid *Rosa pendulina* × *spinosissima* (*Rosa reversa* Waldst. and Kit) Using Biochemical Patterns of the Plant Blossoms

**DOI:** 10.3390/plants12030505

**Published:** 2023-01-22

**Authors:** Nina Kunc, Metka Hudina, Jože Bavcon, Branko Vreš, Zlata Luthar, Kristina Gostinčar, Maja Mikulič-Petkovšek, Gregor Osterc, Blanka Ravnjak

**Affiliations:** 1Department of Agronomy, Biotechnical Faculty, University of Ljubljana, Jamnikarjeva 101, 1000 Ljubljana, Slovenia; 2University Botanic Garden, Biotechnical Faculty, University of Ljubljana, Ižanska Cesta 15, 1000 Ljubljana, Slovenia; 3Jovan Hadži Institute of Biology SRC SASA, Novi Trg 3, 1000 Ljubljana, Slovenia

**Keywords:** autochthonous rose, HPLC-MS, flow cytometry, hybrid, phenols, *Rosa pendulina*, *Rosa spinosissima*, *Rosa pendulina* × *spinosissima*

## Abstract

The *Rosa* genus is characterized by great variability and, consequently, they easily hybridize. The petals of *R. pendulina*, *R. spinosissima* and their hybrid *Rosa pendulina* × *spinosissima*, collected in western Slovenia, were included in the research. We performed morphometric analysis using keys to determine roses and genetic analysis to determine the genome size. The phenolic compound content in petals of all rose flowers was measured by liquid chromatography and mass spectrometry (HPLC-MS). Using flow cytometry, we could confirm the native hybridization process due to the amount of 2C DNA. The value of *R. pendulina* was 1.71 pg, of *R. spinosissima* 1.60 pg and of the hybrid 1.62 pg. The value for the hybrid was close to values measured for parent plants and, at the same time, those values of parent plants significantly differed from each other. Our results showed that the content of phenolic compounds in petals decreased after crossing. We found that the highest total phenolic content (178.9 g/kg FW) was measured in *R. spinossisima*, the lowest content was analyzed for the hybrid (84.36 g/kg FW) and the content for *R. pendulina* was between these two values (110.58 g/kg FW). The content of flavanols and flavonols was lowest in the hybrid petals, whereas the content of anthocyanins was highest in the hybrid petals.

## 1. Introduction

The genus *Rosa* is extremely variable, and therefore, easily hybridized. In Slovenia, due to the very diverse climatic conditions, various rose species successfully grow in the wild. Twenty-six species of wild rose have been morphologically described in the country [1]. Genetic diversity in roses is unparalleled in any other flower crop. Many rose species are grown all around the world. Raymond et al. [2] postulated that roses have probably undergone extremely large reticulate evolution based on this strong interspecific hybridization, introgression and polyploidization. These processes have already been widely observed in the past. Erlanson [3] reported in the early 1930s that many botanists had described numerous spontaneous hybrids occurring in nature among related species of rose. As an indication of these hybridizations, wild individuals of roses often show a considerable amount of aborted pollen. Werlemark [4] described a triploid hybrid originating from spontaneous hybridization between tetraploid *Rosa gallica* and diploid *Rosa arvensis* (Synstylae), and more recent literature has mentioned the existence of natural triploid hybrids between *Rosa gallica* and diploid *Rosa moschata* (Synstylae).

Nevertheless, not all rose species are equally considered to be ready to enter the hybridization process. Gauravi et al. [5] studied the genetic diversity of rose species based on dissimilarity calculation, hierarchical cluster analysis, linear discriminant analysis and principal component analysis. *Rosa banksiae*—*Rosa chinensis* ‘Viridiflora’ and *Rosa banksiae*—*Rosa rubiginosa* showed the highest diversity, while *Rosa brunonii* and *Rosa dumalis* showed the least. Raymond et al. [2] even assumed that only eight to twenty species of roses are expected to contribute to the current complex hybrid varieties of roses, namely *Rosa* × *hybrida* and Chinese rose *R. chinensis* (diploid). These are the main species that later contributed to the extensive hybridization process. Hybrid varieties of tea roses have emerged from these crosses, which are the parents of modern roses with extremely diverse characteristics.

A similar genetic structure (DNA amount, chromosome number) generally enhances the possibility of hybridization. A very useful tool for finding potential suitable parents for forming crosses is flow cytometry. Roberts et al. [6] demonstrated the variability among rose DNA in order to assess the predictability of ploidy levels in relation to DNA amounts. The number of chromosomes in the genus *Rosa* ranged from 2n = 2x = 14 to 2n = 8x = 56, and aneuploidy is rare. Amounts of 2C DNA ranged from 0.78 pg in *Rosa xanthina* Lindl. and *Rosa sericea* Lindl. (2n = 2x = 14) to 2.91 pg in *Rosa canina* L. (2n = 5x = 35). The estimated amounts of 2C DNA ranged from 0.83 pg in *Rosa ecae* (2n = 2x = 14) to 3.99 pg in *Rosa acicularis* (2n = 8x = 56). Differences in 1C values (2C amount of DNA/ploidy values) have been found between taxonomic sections of *Rosa*. Ploidy levels could be assigned with certainty to most species and varieties, but the ploidy of some individuals in section *Caninae* is uncertain due to genomic diversity and aneuploidy. Jowkar et al. [7] studied the diversity of 10 species of Iranian *Rosa* spp. (*Rosa moschata*, *Rosa beggeriana*, *Rosa persica*, *Rosa foetida*, *Rosa hemisphaerica*, *Rosa pimpinellifolia*, *Rosa canina*, *Rosa boissieri*, *Rosa orientalis* and *Rosa pulverulenta*) by chromosome counting and flow cytometry. The number of chromosomes and the level of ploidy of the investigated species ranged from diploid (2n = 2x = 14) to hexaploid (2n = 6x = 42). The average value of 2C of different species showed a level from 0.83 pg in *Rosa persica* to a four times higher value of 3.54 pg in *Rosa pulverulenta*.

The exact description and characterization of hybrids that appear in nature is very important. It is important for botanists to identify potentially new species that appear, but it is also very important for horticulturalists, who are interested in new forms that may have great ornamental potential. Hybrid description is generally based on the morphological description of a plant and its organs, but also on the identification of secondary metabolites accumulating in plant organs, while an exact description can be provided using genetic analysis [8]. In angiospermen, including in roses, the secondary metabolites of flowers are much more useful for plant description than the secondary metabolites in fruits, because the substances in fruits are stronger, as well as being dependent on the fruit-development stage and the method of processing, whereas flower substances are more strongly related to specific species.

Grossi et al. [9] described the content of two flavonol glycosides in the petals of *R. spinosissima*, quercetin-4′-glucoside and kaempferol-4′-*O*-glucoside. Shameh et al. [10] determined eight different phenolic acids (gallic acid, caffeic acid, chlorogenic acid, *p*-coumaric acid, rutin, apigenin, cinnamic acid and quercetin) in the petals of six species of *Rosa* L. Cendrowski et al. [11] focused on the phenolic composition of *Rosa rugosa* petals. Among the eight flavonols detected, quercetin 3,4-*O*-diglucoside was present in the largest amount. The petals were usually high in (+)-catechin and ellagic acid. In addition, four anthocyanins were identified: peonidin 3,5-di-*O*-glucoside, which accounts for approximately 85% of all anthocyanin compounds found, cyanidin 3,5-di-*O*-glucoside, 3-*O*-sophoroside peonidin and 3-*O* peonidin -glucoside. Cunja et al. [12] reported the phenolic profile of petals of four botanical roses (*Rosa canina*, *Rosa glauca*, *Rosa rubuginosa* and *Rosa sempervirens*) and three modern varieties (‘Rosarium Uetersen’, ‘Ulrich Brunner Fils’ and ‘Schwansse’). There were seven different anthocyanins and thrity-one flavonols found in the flowers. The petals contained 14 phenolic acids and their derivatives, 15 flavanols and 20 tannins. The composition of phenolic compounds in terms of content greatly varied among the investigated species and varieties, as well as plant organs.

Extensive research has been carried out on the content of phenolic substances in the petals of roses, but very little is known about the content of substances transmitted by crossbreeding. Our aim was to analyze the content of phenolic compounds in the petals of two *Rosa* species (*R. pendulina* and *R. spinosissima*) in detail and the content of phenolic compounds in their hybrid (*Rosa pendulina* × *spinosissima* described as *R. reversa* Waldstein and Kitaibel in 1812) [13]. All of the rose plants were growing on a relatively small area of just ca. 150 m^2^. These plants are recognized as three different genotypes. Two of them, as already mentioned, were clearly determined as *R. pendulina* and *R. spinosisima*, and the third was not one of the species listed in the determination key of Mala flora of Slovenia [14]. This led to our assumption that it is a hybrid between the two other species. There are several hybrid bushes (up to five) growing on the site, which mostly grow from bunches that are presumably connected to each other, so that the individual bushes are not distinctly separated from each other. We performed a detailed genetic characterization of the species based on phylometry. We focused on how bioactive substances in petals help to characterize hybrids. In addition, flow cytometry based on genome size helped us to confirm the presence of the hybrid, which we had assumed based on morphological properties.

## 2. Results

### 2.1. Morphological Characterization

#### 2.1.1. *Rosa pendulina* L. (syn. *R. alpina*)—Alpine Rose

*Rosa pendulina* is a low, ground bush, from 0.5 to 1 m tall. Stems and branches have a thin, shiny, light brown to red-brown bark, covered with scarce prickles. Leaves are oddly pinnate, composed of seven to nine elongated ovate leaflets, with a doubly serrated margin. Leaflets are glabrous or covered with tiny hairs, dark green on the top side, light green on the bottom, 2 to 4 cm long and 1 to 3 cm wide. Sepals are long, hastate and remain on the fruit. The sepal width ranges from 0.24 cm to 0.37 cm, and the sepal length from 1.4 to 3.3 cm (Figure 1 and Figure 2). Flowers are solitary, pink and from 0.9 to 1.9 cm in diameter. Petal length varies from 1.6 to 2.5 cm (Figure 3 and Figure 4). The fruit is generally hanging, elongated egg-shaped, oval, narrowing towards the top in the shape of a bottle, red, 1.5 to 2.5 cm long; it is frequently covered by glandular hairs and has the remains of long, hastate sepals. The flowering period is from June to July and fruit ripens in September.

Distribution (habitat): forest edges, clearings and light forests, on rocky areas and on the edges of scree slopes from medium mountain to the subalpine zone.

#### 2.1.2. *Rosa spinosissima* L. (syn. *R. pimpinellifolia* L.)—Scotch Rose, Burnet Rose

*Rosa spinosissima* is a low to ground shrub, 20 to 100 cm tall (50 cm on average). The stem is erect, branches are thin, densely covered with thin, straight, needle-like and bristle-like prickles. Leaves are oddly pinnate, 4 to 6 cm long, composed of five to eleven round elliptic, doubly serrated leaflets. Leaflets are predominantly glabrous, 0.5 to 1 cm long; the top side is dark green, the bottom is lighter green. Flowers are solitary, white or pale pink, 2.5 to 5 cm in diameter. Peduncles (in var. *pimpinellifolia*) are glabrous, without glands. Styles are free, much shorter than the stamina. The sepal width ranges from 0.25 cm to 0.42 cm; in length, sepals are very short, with values from 0.9 to 1.6 cm (Figure 1 and Figure 2). Petal width ranges from 1.6 to 2.4 cm, whereas petal length varies from 1.7 to 2.5 cm (Figure 3 and Figure 4).

The fruit is tiny, round, 0.5 to 1.5 cm in diameter; when ripe, it is black-purple. The flowering period is in May and June and fruit ripens from August to September.

Distribution (habitat): rocky slopes and slopes with shrubs in lowlands and sunny and dry edges of hills.

#### 2.1.3. *Rosa pendulina* × *spinosissima* = *Rosa reversa* Waldst. and Kit

*Rosa pendulina* × *spinosissima* is a low, ground bush, 0.5 to 1 m. The bush is fully leafy, from bottom to top. Young shoots are slightly reddish and have no prickles. The prickles on older steams are needle-like. Leaves are oddly pinnate, composed of seven to nine elongated ovate leaflets, with a doubly serrated margin. The petiole is covered with scarce prickles and glands.

Flowers are solitary, pink and 3.5 to 5 cm in diameter. In comparison to *R. pendulina* and *R. spinosissima*, the pedunculus of the hybrid is glandular (like *R. pendulina*) and up to 19 cm long (shorter than the pedunculus of *R. pendulina*). The calyx is without glands (like *R. spinossisima*) and the diameter is the same width as the calyx of both parental species. Sepals are approximately as long as the petals (longer than the sepals of *R. spinosissima* and shorter than *R. pendulina* sepals), hastate and remain on the fruit with widths differing from 0.24 to 0.44 cm and with lengths from 1.2 to 2.5 cm (Figure 1 and Figure 2). Petals are from 1.1 to 1.8 cm wide and from 1.7 to 2.4 cm long (Figure 3 and Figure 4). The fruit is oval. Flowering period is from June to July and the fruit ripens in September.

Distribution (habitat): rocky slopes on a sunny exposition at the edge of forests. Found on Mt. Čaven. According to historic references [15] also present on Mt. Nanos, and at Fridrihstein Castle.

### 2.2. Bioactive Substances of Rose Blossoms

There were 60 different phenolic compounds determined in the analyzed rose petals. The statistically highest total phenolic content, 178.9 g/kg FW, was measured in *R. spinossisima*, and the lowest, 84.36 g/kg FW, was found in *R. pendulina* × *spinosissima*, while the content in *R. pendulina* was in the middle (110.58 g/kg FW) (Figure 5).

There was only one hydroxybenzoic acid (HBA) derivate, gallic acid, present in the analyzed petal material. Its content was significantly higher, 1.02 g/kg FW in *R. pendulina* than in *R. pendulina* × *spinosissima* (0.27 g/kg FW) and the lowest content, 0.12 g/kg FW, was in *R. spinosissima*. From the group of hydroxycinnamic acids (HCA), *p*-coumaric acid hexoside 1 and 3-*p*-coumaroylquinic acid were found. The highest content, 1.13 g/kg FW, of HCA was found in *R. pendulina*, compared with 0.28 g/kg FW in petals of hybrid *R. pendulina* × *spinossisima,* and the lowest content, 0.13 g/kg FW, in petals of *R. spinosissima*. There were eight gallotannins identified in the samples. These were *p*-coumaric acid hexoside 2, digalloylhexoside 1, methyl gallate, digalloylhexoside 2, digalloylquinic acid, trigalloyl hexoside 1, trigalloylhexoside 2 and methyl gallate rutinoside. In all genotypes, the highest content corresponded to trigalloyl hexoside 1. Its content reached 43.78 g/kg FW in *R. pendulina* petals, 31.95 g/kg FW in hybrid petals and 17.17 g/kg FW in *R. spinossisima* petals. Thirteen different ellagitannins were identified in our plant material. The highest content, 41.73 g/kg FW, of ellagitannins was analyzed in *R. spinosissima* petals, and the lowest, 18.17 g/kg FW, was in the petals of the hybrid (Table 1).

Table 2 shows the identified flavanols, flavonols, flavones and dihydrochalcone in the analyzed genotypes. In general, the content of flavonols in rose petals was higher than the content of flavanols. The highest content of flavanols, 14.29 g/kg FW, was measured in petals of *R. spinossisima* and the lowest, 6.79 g/kg FW, in petals of *R. pendulina* × *spinosissima*. The highest content of flavonols, 35.16 g/kg FW, was also found in *R. spinossisima* petals, but, unlike with flavanols, the lowest content, 17.27 g/kg FW, was measured in petals of *R. pendulina*. Quercetin-3-glucuronide was only identified in the petals of the hybrid. There was only one flavone, an apigenin derivate, identified in all genotypes. Dihydrochalcone (phloridzin) was also identified. Its content was highest, 7.89 g/kg FW, in *R. spinosissima* petals and lowest, 2.94 g/kg FW, in *R. pendulina* × *spinosissima*.

There was only one anthocyanin, cyanidin-3-glucoside, detected in the petals of *R. pendulina* and *R. pendulina* × *spinosissima*, (Table 3). In the petals of *R. spinosissima*, as expected based on the white color of the petals, no anthocyanins were detected. The anthocyanin content was slightly higher, 0.24 g/kg FW, in *R. pendulina* × *spinosissima*, than in *R. pendulina* petals, 0.19 g/kg FW. The difference between genotypes was statistically insignificant.

### 2.3. Determination by Flow Cytometry

Based on the obtained results, it can be said that there is no statistically significant difference in 2C DNA amounts between the two parent genotypes (*R. pendulina* and *R. spinosissima*) and the hybrid. The only significant difference was between the two parent genotypes, *R. pendulina* and *R. spinosissima* (Table 4). The 2C DNA amount of *R. pendulina* was 1.71 pg, that of *R. spinosissima* was 1.60 pg and that of the hybrid was 1.62 pg. The value of the hybrid *R. pendulina* × *spinosissima* was closer to the value of *R. spinosissima*, which indicates that the hybrid is genetically closer to this parent genotype.

Figure 6 presents a flow cytometry histogram showing numbers of nuclei per channel of rosa and the internal calibration standard, *Petroselinum crispum*, versus channel numbers, which are linearly proportional to the amount of DNA.

## 3. Discussion

The phenolic compounds in *R. pendulina*, *R. spinosissima* and their hybrid *R. pendulina* × *spinosissima (R. reversa)* were investigated in detail. Rose petals are known for their content of fragrant compounds, used in perfumery and in the food industry. They have also been used in Chinese and European medicine for centuries as ingredients in common cold remedies, because of their content of phenolic compounds [15,16,17]. Additionally, in the research, using morphometry, the differences and similarities between the hybrid and its parental species genotypes were identified. Clear differences were found in the length of the hybrid sepals, with their length being in the middle between the lengths of *R. pendulina* and *R. spinosissima* sepals. While some features of the hybrid, such as the glandular pedunculus and the length of the petals, were more similar to features of *R. pendulina*, the calyx without glands was more similar to that of *R. spinosissima*. When investigating the hybrid and its potential appearance on the territory of Slovenia in the past, we found that the hybrid between *R. pendulina* and *R. spinosissima* was first described in 1812 as *Rosa* × *reversa* Waldst. and Whale., specifically for Hungary. The authors stated that it is a hybrid between the species *R. alpina* (today *R. pendulina*) and *R. spinossisima*. The original name *R. reversa* is without the designation for the hybrid [18]. Host [19] also listed this taxon as a species. It is said to have been present in Hungary, Matrae hill, where it was collected by Kitaibel, and in the subalpine part of Carniola (today part of Slovenia), where it was collected by Franc Hladnik, the founder of the University Botanic Gardens Ljubljana [20]. The hybrid (*R. reversa*) is cited in his manuscripts for Nanos in Botanishe Notizien [21]. He probably sent this information to Host in Vienna, with whom he corresponded [22]. Hladnik mentioned the species *R. alpina* and *R. pimpenifolia* syn. *spinosissima* in other notes [21] for Čaven, but he did not mention their hybrid. After Host, the species *R. reversa* Waldst. and Whale. is mentioned by Fleischmann [15] for the area of Slovenia. He listed three locations: Čaven, Nanos and Fridrihstein Castle (Kočevje). The taxon was later only mentioned by Simkovics [18], and in later works describing the flora of Slovenia, this information disappeared and the species or the hybrid was no longer recorded. In the publication Wild Roses Diversity in Slovenia [22], we only identified the hybrid, with its presence in Čaven being mentioned.

The study revealed a reduction in total phenolic compounds in the petals of the hybrid involved in the experiment. The content of total phenolic compounds in flower petals stayed at 84.36 g/kg FW, behind the content in *R. pendulina*, which had 110.57 g/kg FW and *R. spinosissima* with 118.9 g/kg FW. However, detailed analysis of phenolic compounds showed that the petals of the hybrid species contained quercetin-3-glucuronide, which was only present in the hybrid petals. This substance is therefore a significant identifier of the hybrid. Cendrowski et al. [11] reported that the main polyphenol fraction in *R. rugosa* petals was ellagitannins, constituting from 69 to 74 % of the total polyphenol picture. Among the eight identified flavonols, quercetin 3,4-*O*-diglucoside was present in the highest amount (161 mg/100 g FW). We identified 28 different phenolic compounds. The petals tended to have a high content of (+)-catechin (181 mg/100 g FW) and ellagic acid (49 mg/100 g FW). The total content of phenolics in *R. rugosa* was 107.44 mg/g FW. In all genotypes, the flavonol with the highest amount was quercetin dihexoside. Its content was very low compared to the content of quercetin 3,4-*O*-diglucoside reported by Cendrowski et al. [11]. The highest content of quercetin dihexoside, 12.62 g/kg FW, was measured in our hybrid. Cai et al. [23] studied the phenolic compounds in *R. chinensis*. A total of 36 known and unknown phenolics were identified as hydrolyzable tannins, flavonols and anthocyanins, mainly including gallotannins (mono-, di- or trigalloylglucopyranosides), ellagitannins, quercetin, quercetin/kaempferol mono- and diglycosides, and cyanidin/pelargonidin diglycosides. They found that *R. chinensis* flowers contained a very high level of total phenolics (18.9 g GAE/100 g of DW).

A comparison of the results of our study with those of Cunja et al. [12], who analyzed the phenolic content of *R. canina*, *R. glauca*, *R. sempervirens* and *R. rubiginosa*, shows that Cunja et al. [12] found significantly more anthocyanins than we did. As many as seven were identified, while we identified only one. Cunja et al. [12] found a significantly lower amount of cyanidin-3-glucoside, which we also identified, than we did. It was 26.2 µg/kg FW in *R. glauca* and 31.8 µg/kg FW in *R. rubiginosa*. There were no anthocyanins identified in *R. canina* and *R. sempervirens*. Cendrowski et al. [11] reported four anthocyanins in the petals of *R. rugosa*, cyanidin 3,5-di-*O*-glucoside, peonidin 3-*O*-sophoroside, peonidin 3,5-di-*O*-glucoside and peonidin 3-*O*-glucoside, of which the predominant one, peonidin 3,5-di-*O*-glucoside, represented approx. 85 % of all the determined anthocyanin compounds. Bioelly et al. [24] analyzed the flavonoid metabolism in the petals of more than 100 cyanic cultivars of *Rosa* × *hybrida*. The total anthocyanin content was 60 mg/g DW, consisting of various mixtures of cyanidin 3,5-diglucoside and pelargonidin 3,5-diglucoside. Only pure cyanidin 3,5-diglucoside was found to accumulate in amounts above 60 mg/g. Only small amounts of the related 3-monoglucosides were detected and peonidin 3,5-diglucoside was rarely present.

Sliwinska [25] reported that the size of the rose genome is between 0.25 and 3.05 pg. Bennett and Leitch [26], however, noted that 2C values in *Rosaceae* range from 0.20 to 7.30 pg and Yokoya et al. [27] reported the genome size of *R. persica*, *R. foetida* and *R. canina* to be 0.84, 1.95 and 2.91 pg, respectively. Greilhuber [28] estimated the 2C DNA amount of *R. canina*. He found that the size of the genome was 2.86 pg. Jowkar et al. [7] found that the genome size in *R. persica* was 0.83 pg, in *R. moschata* 1.21 pg, in *R. beggeriana* 1.07 pg, in *R. foetida* 1.91 pg, in *R. hemisphaerica* 2.05 pg, in *R. pimpinellifolia* 1.93 pg, in *R. canina* 2.95 pg, in *R. boissieri* 2.96 pg, in *R. orientalis* 2.94 pg and in *R. pulverulenta* 3.54 pg, which was the largest. Our obtained results for *R. spinosissima* (1.60 pg) are slightly lower than those obtained by Jowkar et al. [7]. In general, there has been very little research conducted on the bars that we included in the research, so we were unable to compare them with other bars. It can be noted that our obtained results (1.71 pg and 1.62 pg) do not match others, although they coincide with those mentioned by Yokoya et al. [7] and Sliwinska [25]. Allum et al. [29] studied the *R. rugosa* hybrid ‘Martin Frobisher’ × ‘Mistress Quickly’. They concluded that it is a diploid (2n = 14), with an average 2C DNA amount of 1.06 pg. He compared the results with Yokoya et al. [27], who studied the diploid *R. rugosa* var. *album*. He found that his obtained value was much higher than in the mentioned research, where they determined a size of 0.98 pg. Differences in flow cytometry estimation occur due to the presence of some secondary metabolites in the cell cytosol, which can cause stoichiometric error in the flow cytometry estimation of nuclear DNA content. There is no completely reliable method that would completely eliminate the effect of these compounds on the nuclei, so using plant parts without staining inhibitors is recommended [25].

However, comparing the results of flow cytometry with the results of analysis of the bioactive substances in the flowers, it can be seen that, despite the number of genomes, the hybrid is more similar to the parent plant *R. spinosissima*, although this is not the case when comparing bioactive substances. The total phenolic content was lower in the hybrid (84.36 g/kg FW) than in the two parent plants (110.58 g/kg FW; 118.9 g/kg FW). The gallic acid content of the hybrid (0.27 g/kg FW) was closer to *R. spinosissima* (0.12 g/kg FW), and the content of ellagitannins and gallotannins was closer to the parent plant *R. pendulina*. When examining flavanols and flavonols, it can be seen that the obtained values of the cross are closer to *R. pendulina*. Anthocyanin cyanidin-3-glucoside was also present in the hybrid (0.24 g/kg FW) and its value was fairly close to the value in the parent plant *R. pendulina* (0.19 g/kg FW).

In summary, we determined the genome sizes of three naturally growing rose species. The process of flow cytometry itself allowed the characterization of the somatic material according to the level of ploidy. The characterization of the ploidy level is of interest for the study of a complex and extensive genus, such as *Rosa*, for which no comprehensive identification key exists. Due to such high variability among studied genotypes, flow cytometry is not always accurate in determining the ploidy level of some genotypes among botanical species [30].

## 4. Materials and Methods

### 4.1. Sampling Locality Description

The area in which the species *R. pendulina*, *R. spinosissima* and their hybrids were sampled is located on the Čaven mountain range, at the beginning of a plateau, the altitude of which is approx. 1100 m above sea level (Figure 7).

The growing area is exposed to strong bora winds. The mountain range is covered with subalpine beech forest with open grassy areas. It is a limestone and dolomitized ridge that steeply descends towards the Vipava Valley. The slope on which roses included in the experiment grow is, in general, strongly exposed to sun, although in some places it is partially shaded by beech trees. There are overgrown grasslands. On the grasslands next to the rose bushes, subalpine species such as *Senecio abrotanifolius* L., *Gentiana lutea* subsp. *symphyandra* L., *Rhododendron hirsutum* L., *Potentilla caulescens* Torn., *Gentiana clusii* Perr. and Song., *Primula auricula* L., *Lilium carniolicum* Bernh. *Daphne cnenorum* L. appear and also *Galanthus nivalis* L., *Narcissus poeticus* subsp. *radiiflorus* (Salisb.) Baker. are present. It is the locality of the endemic plant species *Hladnikia pastinacifolia* Rchb. and some typically Mediterranean species such as *Echinops ritro* subsp. *ruthenicus* (M. Bieb.) *Nyman, Inula hirta* L., *Centaurea rupestris* L., *Iris pallida* subsp. illyrica (Tomm. ex Vis.) K. Richt., *Iris graminea* L., *Iris sibirica* subsp. erirrhiza (Pospichal) T. Wraber., *Genista sericea* Wulf., *Genista holopetala* Fleischm., *Satureja montana* L., *Satureja subspicata* subsp. *liburnica* Šilić. The genotype *R. spinosissima* is present on the edge of the grassland near rocks in the sun. Plants of the genotype *R. pendulina* grow 20 m away, in shade on the edge of the forest and, additionally, there is a location of plants of their hybrid 30 m further away.

### 4.2. Plant Material Sampling

For the analysis of phenolic compounds, the petals of roses *R. pendulina*, *R. spinosissima* and their cross were collected from rose bushes growing in Čaven (western Slovenia) in June 2020, in the BBCH65 phenophase [31]. All of the plants from which we collected the petals were growing in the same climatic conditions, on a total area of 150 m^2^. *R. pendulina* grew in a shady forest edge, *R. spinosissima* over a stone wall where it was exposed to sun and the bora, and hybrids between rocks on the forest edge, where they were exposed to sun for half the day. The plant material of *R. pendulina* and *R. spinosissima* was collected from three bushes each and from three bushes of their hybrid (there were five bushes in fact, but only three of them bloomed (Figure 8). The material was placed on ice and transferred to the laboratory, where the material was stored at −20 °C until further analysis.

For morphological analyses, we harvested, in May 2022, ten flowers from a bush of *R. pendulina* and seven flowers of *R. spinosissima* at the described location. We took a smaller number of flowers from the latter because only so many flowers were properly opened. The remaining flowers had either already bloomed or were still in the bud phase. Then, we took five flowers from each of four clearly separated hybrid bushes, but only three from one, because there were no more present. We put the flowers between two glass plates and pressed them. The pressed flowers were then photographed together with a scale, from the upper and lower sides. The removed flowers were then herbarized.

All flowers of *R. pendulina*, *R. spinosissima* and the hybrids had already been measured at the sampling site for the length and width of the carpel, as well as the length of the flower stalk. With the help of the ImageJ program (Version 1.53t), we measured the length of the sepals, the width of the sepals at the base of the inflorescence, the length of the petals and the width of the petals at their widest part. Morphological differences and similarities of hybrids and the parental species were described at the same time.

Fresh young leaves of the analyzed plants were used to determine the content of nuclear DNA. From each of the mentioned bushes included in the research, we collected three leaves representing three repetitions.

### 4.3. Extraction and Analysis of Phenolic Compounds

Extraction of rose petals was performed according to the extraction method as described by Kunc et al. [32]. The analyses were performed in triplicate. The harvested flowers from native grown plants (see Section 4.2) were divided into three replicates with three flowers. Each sample was ground with liquid nitrogen in a mortar and the measured mass of the sample was placed in a centrifuge tube and an extraction solution (3% formic acid in methanol with bidistilled water) was added. The ratio of weighed sample to extraction solution was 1:5. The weight of the samples was 0.02 g and the volume of the extraction solution was 1 mL. Extraction was then carried out in a chilled ultrasonic bath (Iskra PIO, SONIS 4 GT) on ice for 1 h, after which the extract was centrifuged (Eppendorf centrifuge 5810 R) at 10,000× *g* for 7 min at 4 °C. Supernatants were filtered through a 0.20 µm polyamide/nylon filter (Macherey-Nagel, Dϋren, Germany). The vials with the extracts were kept at −20 °C until further analysis of the phenolic compounds.

Analysis of phenolic components was performed on a Thermo Scientific Dionex HPLC system with a diode array detector (Thermo Scientific, San Jose, CA, USA) connected to Chromeleon workstation software. The chromatographic method for phenolic analysis has previously been described by Mikulic-Petkovsek et al. [33]. The detector was set to three wavelengths: 280 nm, 350 nm and 530 nm. The mobile phases were phase A: 3% acetonitrile/0.1% formic acid/96.9% double distilled water; phase B: 3% water/0.1% formic acid/96.9% acetonitrile. The gradient elution of the two mobile phases is described in Mikulic-Petkovsek et al. [34] and their flow rate was 0.6 mL/min. The column used was a Gemini C18 (150 × 4.6 mm 3 μm; Phenomenex, Torrance, CA, USA), heated to 25 °C.

Phenolic compounds were identified by mass spectrometer (LTQ XL Linear Ion Trap Mass Spectrometer, Thermo Fisher Scientific, Waltham, MA, USA) with electrospray ionization (ESI) operating in positive (anthocyanins) or negative (all other phenols) ionization mode. All conditions of the mass spectrometer were set as reported by Mikulic-Petkovsek et al. [33]. Spectral data were generated with Excalibur (Thermo Scientific) software. Identification of the compounds was confirmed by comparing retention times and their spectra, by adding a standard solution to the sample, and by fragmentation and comparison with literature data.

The contents of phenolic compounds were calculated from the areas of the sample peaks and the corresponding standards. The content was expressed as g/kg fresh weight (FW) [32].

### 4.4. Flow Cytometry

The content of nuclear DNA in rose leaves was measured by flow cytometry (FCM) according to the method described by Doležel et al. [35] using fresh parsley leaves (*Petroselinum crispum* (P. Mill.) Nyman ex A.W. Hillleaf) (2C amount DNA = 4.5 pg) as internal standard. We adapted the described method according to the material we used and according to what proved to be the optimal analysis process during the experiments themselves. Analyses were performed in triplicate. A small portion of fresh rose and parsley leaves were chopped together in a Petri dish with 1 mL of Galbraith’s buffer supplemented with 5 mM sodium metabisulfite and 1% polyvinylpyrrolidone 10000 [36]. The homogenate was filtered through a 30 µm nylon mesh into a labeled sample tube. Propidium iodide (50 µg/mL), RNase (50 µg/mL) and 1 mL of Galbraith’s buffer were then added. The sample was analyzed using a flow cytometer CyFlow space (Sysmex Partec, GmbH). Fluorescence in at least 7000 nuclei was measured for each sample.

The size of the genomes was calculated from the peak areas of the sample and the corresponding standard according to the equation described by Doležel et al. [35].

### 4.5. Statistical Analysis

Results were analyzed with the statistical program R commander using one-way analysis of variance (ANOVA) and with Microsoft Excel 2016. Duncan’s test was used for flow cytometric analyzes and for anthocyanin analysis, to compare treatments when ANOVA showed significant differences between values. Tukey test was used to analyze phenolic compounds, to compare treatments when ANOVA showed significant differences between values. Results are given as mean value with standard error (SE). When *p*-values were less than or equal to 0.05, the differences among genotypes were statistically significant.

The morphometric data were analyzed using the Statistica 8.0 program to assess the potential differences among the two parent species and their hybrid. Box-plot diagrams were made for each variable.

## 5. Conclusions

It can be concluded that the petals of the hybrid *R. pendulina* × *spinosissima* collected in 2020 in western Slovenia have a lower total phenolic content than the parent plants. Our results show that, with crossing, the content of phenolic compounds in petals decreased. Looking at the individual groups of phenolic compounds, it can be concluded that the content of all phenolic compounds in the hybrid is lower. The content of anthocyanins was only expected in *R. pendulina* and in the hybrid. Its content was higher in the hybrid, so it can be concluded that hybridization enhances anthocyanin accumulation. In general, it should be noted that very little research has been carried out on how the content of phenolic substances changes with crossbreeding. We did not find any research that mentioned the content of phenolic compounds in the petals of the plants we included in the study. Using flow cytometry, we found that there were statistical differences between the genome size of *R. spinosissima* and *R. pendulina*, and their hybrid was closer to the parent *R. spinosissima* in terms of genome size, despite being closer to *R. pendulina* in terms of phenolic composition. It should also be mentioned that the hybrid differs from both parents in that the young shoots are without spines. Sepals are about as long as sepals, longer than *R. spinosissima* sepals and shorter than those of *R. pendulina*. A difference in the shape of the fetus can also be detected.

## Figures and Tables

**Figure 1 plants-12-00505-f001:**
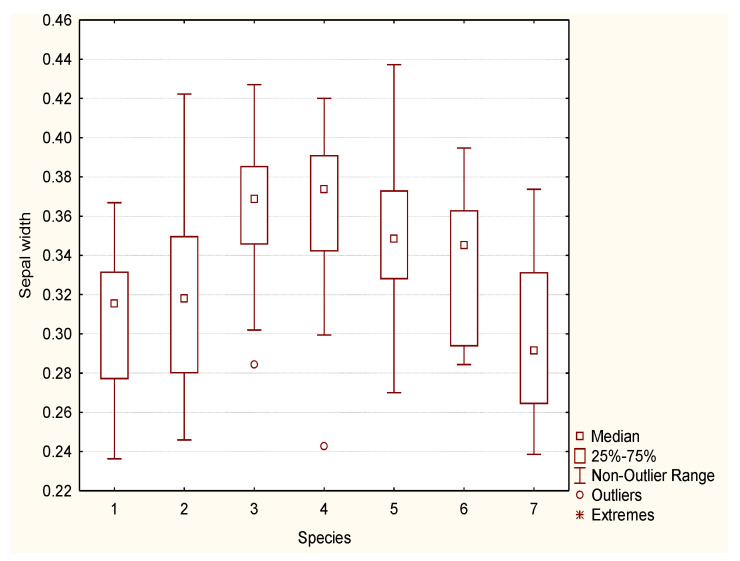
Sepal width (cm) box-plot diagram of *R. pendulina* (1), *R. spinosissima* (2) and their hybrid (*R. reversa*) (3–7).

**Figure 2 plants-12-00505-f002:**
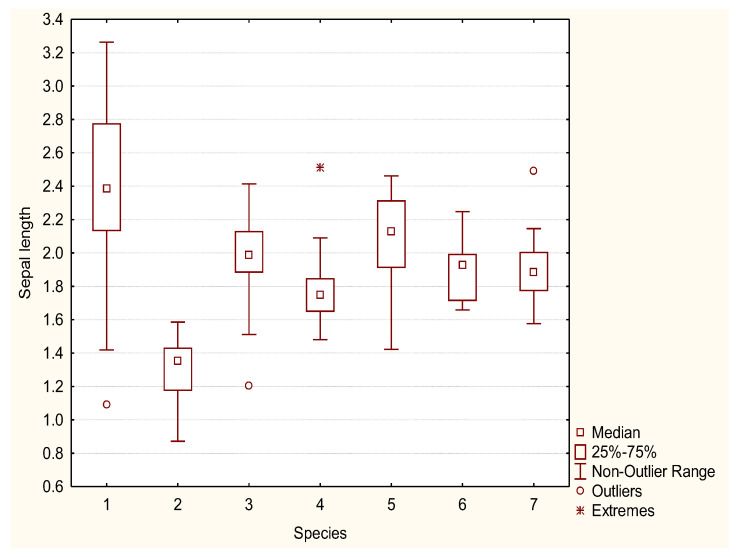
Sepal length (cm) box-plot diagram of *R. pendulina* (1), *R. spinosissima* (2) and their hybrid (*R. reversa*) (3–7).

**Figure 3 plants-12-00505-f003:**
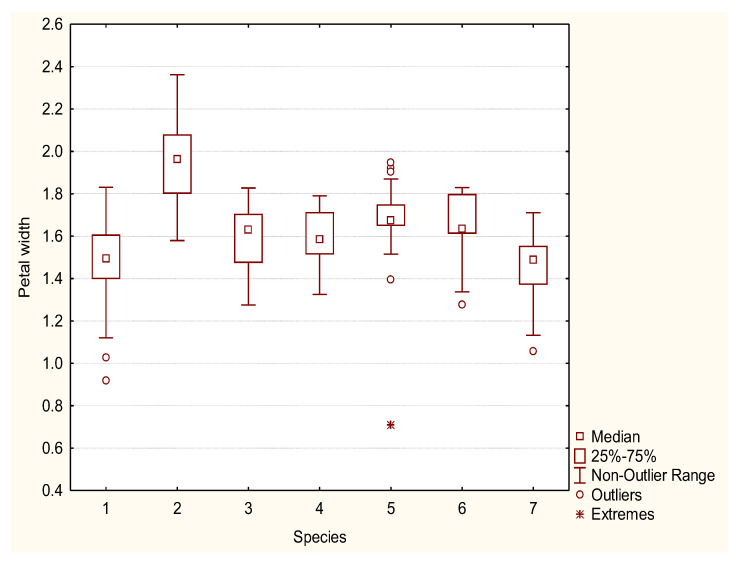
Petal width (cm) box-plot diagram of *R. pendulina* (1), *R. spinosissima* (2) and their hybrid (*R. reversa*) (3–7).

**Figure 4 plants-12-00505-f004:**
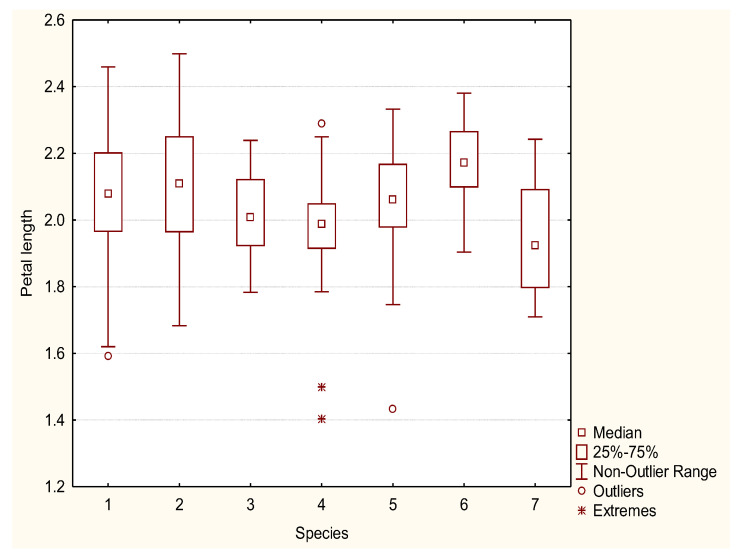
Petal length (cm) box-plot diagram of *R. pendulina* (1), *R. spinosissima* (2) and their hybrid (*R. reversa*) (3–7).

**Figure 5 plants-12-00505-f005:**
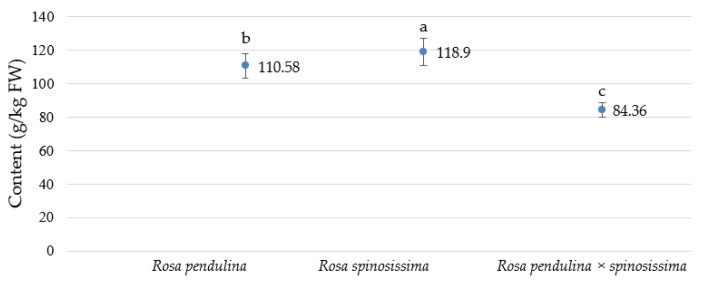
Comparison of total analyzed phenolic content (g/kg FW) ± standard error in petals of *R. pendulina*, *R. spinosissima* and the hybrid *R. pendulina* × *spinosissima* in the region of west Slovenia in 2020. The Tukey test was used to compare treatments when ANOVA showed significant differences among values (*p* < 0.05). Different letters indicate significant differences between genotypes.

**Figure 6 plants-12-00505-f006:**
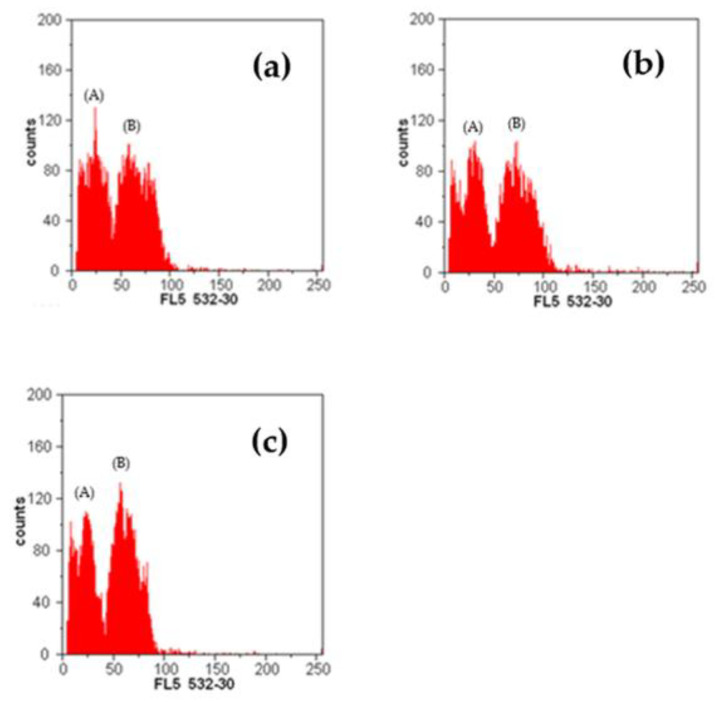
Flow cytometry histogram showing numbers of nuclei per channel of rosa and the internal calibration standard, *Petroselinum crispum*, versus channel numbers (which are linearly proportional to the amount of DNA). (**a**) G1 peaks of *Rosa spinosissima* (A) and standard (B). (**b**) G1 peaks of *Rosa pendulina* (A) and standard (B). (**c**) G1 peaks of *Rosa pendulina* × *spinosissima* (A) and standard (B).

**Figure 7 plants-12-00505-f007:**
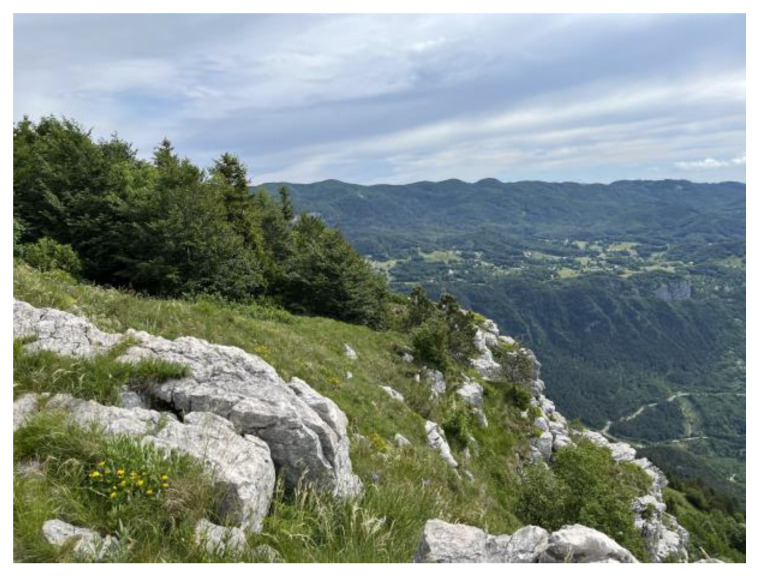
Čaven mountain range.

**Figure 8 plants-12-00505-f008:**
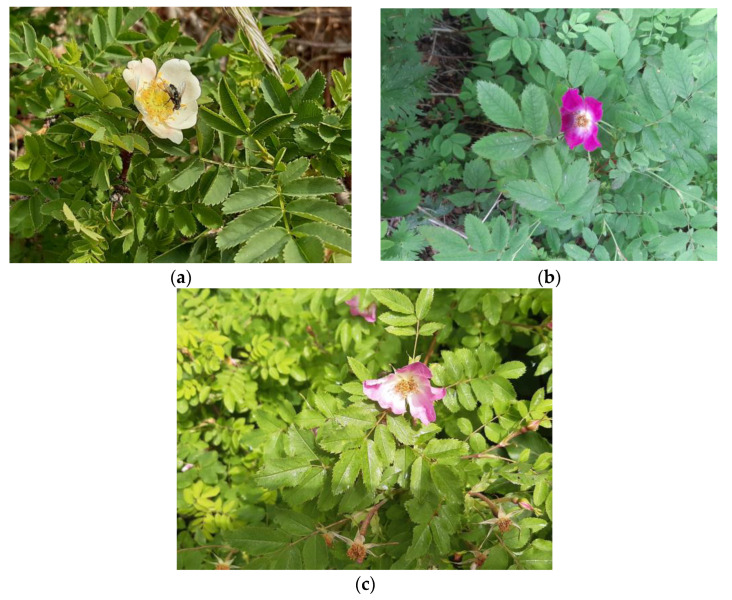
(**a**) *R. spinosissima*, (**b**) *R. pendulina* and (**c**) *R. pendulina* × *spinosissima*.

**Table 1 plants-12-00505-t001:** The content ± standard error (g/kg FW) of phenolic compounds (HBA, HCA, gallotannins and ellagitannins) in petals of *R. pendulina*, *R. spinosissima* and the hybrid *R. pendulina* × *spinosissima* in the region of west Slovenia in 2020. Different letters indicate significant differences between genotypes.

Phenolic Group	Compound	*Rosa pendulina*	*Rosa spinossisima*	*Rosa pendulina × spinosissima*
Hydroxybenzoic acid (HBA)	gallic acid	1.02 ± 0.04 a	0.12 ± 0.05 c	0.27 ± 0.016 b
Hydroxycinnamic acid derivatives (HCA)	*p*-coumaric acid hexoside 1	0.002 ± 0.0 c	0.007 ± 0.0 a	0.004 ± 0.0 b
	3-*p*-coumaroylquinic acid	0.007 ± 0.0 a	0.003 ± 0.0 c	0.005 ± 0.0 b
	TOTAL	1.029 ± 0.04 a	0.13 ± 0.00 c	0.279 ± 0.0 b
Gallotannins	*p*-coumaric acid hexoside 2	0.68 ± 0.03 b	1.13 ± 0.07 a	1.02 ± 0.09 a
	Digalloylhexoside 1	1.02 ± 0.04 a	0.12 ± 0.005 c	0.27 ± 0.016 b
	Methyl gallate	0.19 ± 0.04 c	0.74 ± 0.06 a	0.43 ± 0.05 b
	Digalloylhexoside 2	1.31 ± 0.04 a	0.21 ± 0.04 b	0.26 ± 0.04 b
	Digalloylquinic acid	0.01 ± 0.0 a	0.03 ± 0.0 c	0.02 ± 0.0 b
	Trigalloyl hexoside 1	43.78 ± 3.4 a	17.17 ± 0.7 c	31.95 ± 1.8 b
	Trigalloylhexoside 2	0.008 ± 0.0 b	0.01 ± 0.0 a	0.01 ± 0.0 ab
	Methyl gallate rutinoside	0.09 ± 0.01 b	0.14 ± 0.004 a	0.05 ± 0.003 c
	TOTAL	47.09 ± 3.56 a	19.55 ± 0.879 c	34.01 ± 1.999 b
Ellagitannins	Di HHDP hexoside 1	0.11 ± 0.004 a	0.36 ± 0.0 c	0.09 ± 0.0 b
	Di HHDP hexoside 2	0.36 ± 0.01 b	0.71 ± 0.02 a	0.28 ± 0.01 c
	HHDP digalloylhexoside 1	0.35 ± 0.03 c	0.94 ± 0.02 a	0.52 ± 0.0 b
	HHDP galloylhexoside	0.39 ± 0.03 a	0.15 ± 0.02 c	0.28 ± 0.01 b
	HHDP digalloyhexoside 2	0.41 ± 0.09 b	0.68 ± 0.03 a	0.5 ± 0.01 ab
	Galloyl bis HHDP hexoside 1	1.01 ± 0.03 b	2.37 ± 0.29 a	0.89 ± 0.02 b
	Galloyl bis HHDP hexoside 2	5.14 ± 0.26 a	8.55 ± 0.53 b	7.7 ± 0.65 b
	Galloyl bis HHDP hexoside 3	4.56 ± 0.47 a	3.88 ± 0.22 a	1.5 ± 0.04 b
	Trigalloyl HHDP hexoside	1.2 ± 0.14 b	2.3 ± 0.39 a	0.57 ± 0.03 b
	Vescalagin 1	1.73 ± 0.14 a	1.96 ± 0.06 a	1.2 ± 0.06 b
	Vescalagin 2	5.3 ± 0.16 a	5.4 ± 3.04 a	1.5 ± 0.13 a
	Vescalagin 3	0.57 ± 0.07 b	1.09 ± 0.19 a	0.27 ± 0.02 b
	Vescalagin 4	8.5 ± 0.32 b	12.99 ± 0.29 a	3.87 ± 0.18 c
	TOTAL	29.63 ± 1.754 b	41.73 ± 5.1 a	18.17 ± 1.16 c

Note:The Tukey test was used to compare treatments when ANOVA showed significant differences among values (α = 0.05).

**Table 2 plants-12-00505-t002:** Contents ± standard error (g/kg FW) of flavanols, flavonols, flavones and dihydrochalcone in petals of *R. pendulina*, *R. spinosissima* and the hybrid *R. pendulina* × *spinosissima* in the region of west Slovenia in 2020. Different letters indicate significant differences between genotypes.

Phenolic Group	Compound	*Rosa pendulina*	*Rosa spinossisima*	*Rosa pendulina × spinosissima*
Flavanols	Dimer PA monogallate 1	1.68 ± 0.06 b	3.93 ± 0.48 a	1.48 ± 0.02 b
	Dimer PA monogallate 2	2.96 ± 0.2 a	3.37 ± 0.1 a	2.06 ± 0.21 b
	Dimer PA monogallate 3	2.75 ± 0.1 b	4.2 ± 0.12 a	1.25 ± 0.06 c
	Procyanidin dimer 1	0.32 ± 0.01 b	0.62 ± 0.02 a	0.24 ± 0.01 b
	Procyanidin dimer 2	0.66 ± 0.05 c	1.74 ± 0.05 a	0.97 ± 0.01 b
	Catechin	0.6 ± 0.04 a	0.24 ± 0.04 c	0.44 ± 0.04 b
	Procyanidin trimer	0.49 ± 0.04 a	0.19 ± 0.01 c	0.35 ± 0.02 b
	TOTAL	9.41 ± 0.5 b	14.29 ± 0.82 a	6.79 ± 0.37 c
Flavonols	Quercetin rhamnosyl dihexoside	0.09 ± 0.01 a	0.02 ± 0.002 b	0.02 ± 0.0 b
	Quercetin dihexoside	5.21 ± 0.03 c	11.17 ± 0.38 b	12.62 ± 0.02 a
	Quercetin galloyl hexoside 1	1.34 ± 0.07 a	1.19 ± 0.0 a	1.2 ± 0.09 a
	Kaempferol dihexoside	1.93 ± 0.1 a	1.68 ± 0.01 a	1.71 ± 0.12 a
	Quercetin-3-rutinoside	0.18 ± 0.02 b	0.29 ± 0.004 a	0.04 ± 0.002 c
	Quercetin galloyl hexoside 2	0.73 ± 0.08 b	1.2 ± 0.02 a	0.18 ± 0.01 c
	Quercetin pentoside hexoside	1.43 ± 0.16 b	2.27 ± 0.03 a	0.34 ± 0.02 c
	Quercetin-3-galactoside	1.67 ± 0.12 b	2.22 ± 0.02 a	0.42 ± 0.02 c
	Quercetin-3-glucoside	1.44 ± 0.03 b	1.8 ± 0.05 a	0.57 ± 0.10 c
	Quercetin-3-xyloside	0.02 ± 0.001 a	0.01 ± 0.001 b	0.012 ± 0.001 b
	Kaempferol rhamnosyl hexoside 1	0.3 ± 0.007 a	0.15 ± 0.01 b	0.17 ± 0.01 b
	Quercetin arabinopyranoside	0.56 ± 0.02 c	4.93 ± 0.12 a	1.22 ± 0.08 b
	Kaempferol hexoside	0.04 ± 0.001 c	0.36 ± 0.009 a	0.09 ± 0.005 b
	Isorhamnetin-3-glucuronide	0.06 ± 0.01 b	0.3 ± 0.01 a	0.03 ± 0.001 a
	Quercetin-3-glucuronide	-	-	0.9 ± 0.07
	Quercetin-3-arabinofuranoside	0.19 ± 0.02 c	1.62 ± 0.03 a	0.31 ± 0.02 b
	Quercetin acetylhexoside	0.006 ± 0.001 c	0.05 ± 0.001 a	0.01 ± 0.001 b
	Quercetin-3-rhamnoside	0.61 ± 0.07 c	5.25 ± 0.10 a	1.02 ± 0.06 b
	Kaempferol-3-glucuronide	0.75 ± 0.04 a	0.18 ± 0.01 c	0.42 ± 0.02 b
	Quercetin galloylpentoside 1	0.03 ± 0.002 a	0.01 ± 0.0 c	0.02 ± 0.001 b
	Quercetin galloylpentoside 2	0.01 ± 0.001 b	0.02 ± 0.001 a	0.01 ± 0.001 b
	Kaempferol pentoside 1	0.14 ± 0.002 b	0.21 ± 0.01 a	0.14 ± 0.01 b
	Kaempferol-3-rhamnoside	0.10 ± 0.004 b	0.03 ± 0.001 c	0.29 ± 0.02 a
	Kaempferol rhamnosyl dihexoside	0.01 ± 0.002 c	0.03 ± 0.001 b	0.08 ± 0.006 a
	Kaempferol galloyl pentoside	0.007 ± 0.04 c	0.05 ± 0.002 a	0.02 ± 0.001 b
	Quercetin rhamnosyl hexoside 1	0.28 ± 0.03 a	0.04 ± 0.02 b	0.02 ± 0.0 b
	Quercetin rhamnosyl hexoside 2	0.12 ± 0.0 a	0.04 ± 0.0 b	0.008 ± 0.0 b
	Kaempferol rhammnosyl hexoside 2	0.12 ± 0.02 a	0.037 ± 0.002 b	0.008 ± 0.0 b
	TOTAL	17.27 ± 0.891 c	35.16 ± 0.862 a	21.88 ± 0.689 b
Flavones	apigenin derivate	0.02 ± 0.001 a	0.02 ± 0.001 a	0.02 ± 0.001 a
Dihydrochalcone	phloridzin	5.13 ± 0.46 b	7.89 ± 0.22 a	2.94 ± 0.16 c

-: Substance was not detected. The Tukey test was used to compare treatments when ANOVA showed significant differences among values (α = 0.05).

**Table 3 plants-12-00505-t003:** The average content ± standard error (g/kg FW) of cyanidin-3-glucoside in petals of *R. pendulina* and *R. pendulina* × *spinosissima* in the region of western Slovenia in 2020. Mean values with corresponded standard error presented. Letters denote a significant difference among values.

Phenolic Group	Compound	*Rosa pendulina*	*Rosa pendulina* × *spinosissima*
anthocyanins	cyanidin-3-glucoside	0.19 ± 0.02 a	0.24 ± 0.01 a

Note: The Duncan test was used to compare treatments when ANOVA showed significant differences among values (α = 0.05).

**Table 4 plants-12-00505-t004:** 2C DNA amounts (pg ± SE) of rosa *R. pendulina*, *R. spinosissima* and *R. pendulina* × *spinosissima* measured by flow cytometry (all estimates based on three replicates). Different letters indicate significant differences between genotypes.

Species	2C DNA Amount
*R. pendulina*	1.71 ± 0.04 a
*R. spinosissima*	1.60 ± 0.006 b
*R. pendulina* × *spinosissima*	1.62 ± 0.01 ab

Note: The Duncan test was used to compare treatments when ANOVA showed significant differences among values (α = 0.05).

## Data Availability

All data are presented within the article.

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
