# Peer review of "Characterization of the Slovene Autochthonous Rose Hybrid Rosa pendulina × spinosissima (Rosa reversa Waldst. and Kit) Using Biochemical Patterns of the Plant Blossoms"

_plants, 2023, doi:10.3390/plants12030505_

Round 1
Reviewer 1 Report
The studies indicated in the article are undoubtedly relevant. The exact description and characterization of hybrids that appear in nature is very important. It is important for botanists to identify potentially new species that appear, but it is also very important for horticulturalists, who are interested in new forms that may have great ornamental potential. The authors carried out an interesting integrative study of 3 taxa of the genus Rosa: two species R. pendulina and R. spinosissima and one hybrid R. pendulina × spinosissima. Morphological features, genome size, as well as the composition and content of phenolic compounds in rose petals were studied. The three taxa differ in the studied characters. The most interesting thing is that the hybrid contains the least total phenolic content, flavanols and flavonols, whereas the content of anthocyanins was highest in the hybrid petals, compared to the parent species. The authors made a good overview of the research topic. It is very important that a marker compound for the hybrid, quercetin-3-glucuronide, was identified, which was not identified in any of the parental species. The composition of the manuscript is correct. However, several issues need clarification and explanation.
1. In Figure 5, it is desirable to indicate with what statistical test the difference between the content of phenolic compounds was analyzed and what was the confidence interval.
2. Line 191. R. pendulina × spinosissima (1.27 g/kg FW) - probably a typo? Table 1 shows 0.27 g/kg FW in hybrid petals.
3. Table 1 – 4, it is desirable to indicate with what statistical test the difference between the content of phenolic compounds was analyzed and what was the confidence interval. And transfer this information from the table name to the notes. In the title of Table 1, replace Rosa spinosissima with R. spinosissima.
4. In section 4.2. Plant material sampling shows that, «R. pendulina grew in a shady forest edge, R. spinosissima over a stone wall where it was exposed to sun and the bora, and hybrids between rocks on the forest edge, where they were exposed to sun for half the day». Three taxa were collected in different places according to the light level. R. spinosissima was in the sun, R. pendulina was in the shade, and the hybrid was exposed to solar insolation for about half a day. Secondary metabolites, especially phenolic compounds, strongly depend on ecological and geographical factors and, first of all, on the amount of solar insolation received by plants. Could the difference in the content of the studied substances depend on where the plant grew, in the shade or in the sun? Do the authors suggest such a version of the explanation of the difference in the content of substances between species?
5. It would also be nice to discuss how other plant species, not only plants of the genus Rosa, accumulate phenolic compounds. Are hybrids of other plant species losing phenolic content or not? This is not discussed in the article.
6. In section 4.2. Plant material sampling, line 386. In what phenophase were the plants collected? What is the phenophase BBCH65?
7. Figure 8. It is advisable to highlight the name of the species in italics and designate figures A, B, C according to the rules of the journal.
8. Which leaves were taken for genome size analysis? Fresh leaves or dried in silica gel?
9. In conclusion, the phrase "cruciferous vegetables" does not quite fit the context.
10. In conclusion, it is also necessary to mention morphological features. How did the hybrids differ from the parental genotypes in terms of morphometric characteristics? To what species is the hybrid closer morphologically?
Reviewer 2 Report
Paper:
Characterization of the Slovene autochthonous rose hybrid Rosa pendulina x spinosissima (Rosa reversa Waldst. & Kit) using biochemical patterns of the plant blossoms
Authors: Kunc et al.
The paper is, in my opinion, well written and interesting, but I noticed many typos; accordingly, I suggest that you read the paper again carefully. I also suggest to better explain how you checked the ANOVA assumptions (independence, normality, homogeneity).
Therefore, I recommend the paper for publication in Plants after it has been thoroughly revised. I provide suggestions and specific comments for revision below.
1. Line 149: What do you mean by (20) 50 (100) cm tall?
2. Line 185: On the y-axis of Figure 5, correct "Conent" to "Content".
3. Line 186: Probably you calculate the standard error of the mean? Why are all standard errors the same for each parameter?
4. Line 188: Add the name of the post hoc test and the alfa level below each table.
5. Table 1 and 2: Why is the standard error the same for all three groups? Compare your standard error with other studies for at least some parameters
6. Line 227: In scientific texts, the word significant is usually synonymous with "statistically significant" Not significant means the opposite, i.e., "statistically not significant" In contrast, "not significant" usually implies insignificance, without statistical connotations.
7. Explain why you have indicated SD and not SEM in Table 3.
8. Error in Table 2 (see below):
Quercetin dihexoside 5.21 ± 0.31 c 11.17 ± 0.31 b 12.62 0.31 a
9. Line 390: You claimed that the plant material of R. pendulina and R. spinosissima was collected from one shrub each. You treated multiple petals from the same plant as replicates, what we call pseudo-replications in statistics, one of the ANOVA assumptions is independence. How do you deal with that, explain that? How do you test other ANOVA assumptions (no outliers, homogeneity, normality)?
10. In each table, report the number of replications for each group.
11. Include any statistical tests as a supplement.
12. Line 391: I do not quite understand, how many bushes of your hybrids do you have (3 or 5)?
13. Line 466: Correct "if the p-values were less than or equal to 0.05".
14. I noticed in the paper (also in the references) the wrong use of a hyphen (for example –79, 73–79).
Round 2
Reviewer 2 Report
I am unable to accept the work in this form because there are no true replicates in your study and thus no statistical conclusions (inferental statistics) are possible. You must have more bushes of each variety. With this data you can only do descreptive statistics, I wonder if that is enough for this journal? Of course, you have to explain why you took so small sample. Another option would be not to separate the varieties by statistical analysis and present the results in other ways.
I suggest to read the following paper: https://www.jstor.org/stable/1942661?seq=13#metadata_info_tab_contents
Author Response
"Please see the attachment."

Round 3
Reviewer 2 Report
The paper is ready for publication.